# Ultra-Sensitive and Semi-Quantitative Vertical Flow Assay for the Rapid Detection of Interleukin-6 in Inflammatory Diseases

**DOI:** 10.3390/bios12090756

**Published:** 2022-09-14

**Authors:** Rongwei Lei, Hufsa Arain, Maryam Obaid, Nivriti Sabhnani, Chandra Mohan

**Affiliations:** Department of Biomedical Engineering, University of Houston, Houston, TX 77004, USA

**Keywords:** inflammation diseases, rheumatoid arthritis, interleukin-6, vertical flow assay, ultra-sensitivity, serology, gold nanoparticle (GNP)

## Abstract

The inflammation biomarker Interleukin 6 (IL-6) exhibits a concentration of less than 7 pg/mL in healthy serum but increases 10–100-fold when inflammation occurs. Increased serum IL-6 has been reported in chronic diseases such as rheumatoid arthritis (RA), as well as in life-threatening acute illnesses such as sepsis and cytokine release syndrome (CRS). This work seeks to meet the demand for rapid detection of serum IL-6 both for rapid monitoring of chronic diseases and for triaging patients with acute illnesses. Following the optimization of several types of gold nanoparticles, membrane pore sizes, and buffer systems, an ultra-sensitive vertical flow assay (VFA) was engineered, allowing the detection of recombinant IL-6 in spiked buffer with a limit of detection (LoD) of 10 pg/mL and a reportable range of 10–10,000 pg/mL with a 15-min assay time. The detection of IL-6 in spiked pooled healthy serum exhibited an LoD of 3.2 pg/mL and a reportable range of 10–10,000 pg/mL. The VFA’s stability was demonstrated over 1-day, two-week, four-week, and six-week storage durations at room temperature. The inter-operator CV and intra-operator CV were determined to be 14.3% and 15.2%, respectively. Three reference zones, high, low, and blank, were introduced into the cartridge to facilitate on-site semi-quantitative measurements across a 6-point semi-quantitative range. Finally, the performance of the IL-6 VFA was validated using 20 RA and 20 healthy control (HC) clinical serum samples, using ELISA as the gold standard platform. The ultra-sensitive, rapid IL-6 VFA could potentially be used to triage patients for intensive care, treatment adjustments, or for monitoring disease activity in inflammatory conditions.

## 1. Introduction

Interleukin-6 (IL-6) is a pleiotropic cytokine identified in 1986 as a key functional molecule in adaptive immunity. It then came to be recognized for its impact on T-cell activation and proliferation and B-cell differentiation [1]. The neutralization of IL-6 with antibodies specific to the cytokine or the α-chain receptor improves inflammation in pre-clinical models [2]. Additionally, during acute phase response, various proteins increase 10–100-fold, and IL-6 was found to be a key inducer of this response. It was also noted that IL-6 levels increase rapidly after infection, as seen upon injecting lipopolysaccharide in mice or following acute viral or bacterial infections in humans [3]. Elevated IL-6 levels are present in several pathological states. For instance, IL-6 levels are significantly elevated in obese patients with uncontrolled type 2 diabetes mellitus and are even predictive of disease severity [4]. Compared to healthy controls (HC), IL-6 levels are also significantly higher in patients with various malignancies, including pancreatic cancer, ovarian cancer, renal cell carcinoma, and lung cancer [5,6,7,8,9,10,11,12]. Other diseases with elevated IL-6 levels are listed in Appendix A [5,13,14,15,16,17,18,19,20,21,22,23,24,25,26,27,28,29].

Due to its implication in the pathogenesis of many diseases, IL-6 has been targeted for diagnostics and therapeutics. Cytokine release syndrome (CRS) and sepsis are life-threatening complications following infection and other primary triggers. Elevated levels of IL-6 have been reported as a biomarker for COVID-19-induced CRS, chimeric antigen receptor (CAR)-T therapy-induced CRS, and sepsis patients with worsening disease status [14,22,24]. Patients with worsening CRS may need ICU care and stronger immunosuppressant therapy. With early medical intervention, one can minimize ICU admissions and high dosages of immunosuppressants, which could potentially be detrimental. Takayasu arteritis and giant cell arteritis (GCA), chronic and potentially life-threatening inflammatory diseases of the blood vessels, are further examples where IL-6 plays a pivotal role in disease progression [30]. Importantly, tocilizumab, a humanized antibody to IL-6R, has been shown to improve disease signs and symptoms and sustain remission [30]. In all of the above diseases, irreversible body system damage can occur when the diagnosis is delayed. Rapid detection methods allow for sooner initiation of treatment, which could dramatically improve clinical outcomes.

Rheumatoid arthritis (RA) is a chronic inflammatory autoimmune disease that can lead to joint destruction. This disease is characterized by the release of several pro-inflammatory cytokines, including IL-1, TNFα, IL-6, IL-15, IL-17, and IL-18 [31,32,33,34]. IL-6 levels are linked to the onset of RA as they are expressed in abundance in the synovial fluid of RA patients and are correlated to disease activity and joint destruction [35]. Due to the vital role IL-6 plays in the development of RA, its blockade has become a therapeutic option. Various clinical trials worldwide have reported the anti-IL-6 drug tocilizumab’s efficacy in subduing disease activity, and as a result, it is currently being used in more than 100 countries as a first-line drug for RA treatment [36]. Thus, assaying serum IL-6 levels for identifying patients for therapy and monitoring IL-6 levels to assess disease activity and treatment response in RA presents a much-needed opportunity for rapid testing.

There are many ways to detect IL-6. A common way is an enzyme-linked immunosorbent assay (ELISA) [37], wherein a varying range of IL-6 levels (Appendix A) can be detected. However, ELISA requires two days to complete, which precludes it from being used as a rapid test for routine monitoring at the point of care. There are ELISA kits with shorter durations that can perform the IL-6 assay between 90 min and 3 h; however, their range of detection usually extends from 7.8 to 500 pg/mL. Their analyte detection ranges tend to increase with assay time. Some ELISAs may last 4.5 h or longer. Additionally, they have lower sensitivities.

Another assay platform that has been used for IL-6 detection is a lateral flow assay (LFA), which is more amenable for point-of-care testing. A sandwich immunofluorescent LFA has been reported for detecting IL-6. This IL-6 LFA test had a LoD of 0.38 ng/mL and a linear range between 1.25 ng/mL and 9000 ng/mL with a 20-min assay time [38]. This range and relatively rapid detection provide LFA with an advantage over ELISA and render it a suitable point-of-care detection method. There has been significant experimentation with the LFA technology, using fluorometric, colorimetric, and SERS labels as reporters. Other groups have incorporated gold nanoparticles (GNPs) selected for their higher affinity for detection and the possibility of naked-eye colorimetric signal detection. Paper-based lateral flow assays have demonstrated their effectiveness in detecting analytes in a wide range of clinical samples. However, the small volume capacity of such assays has limited the sensitivity, especially for low-level analytes. Clinical samples with high viscosity typically require dilution to facilitate the flow, further diluting the already low levels of each analyte, thereby intensifying the difficulty of detecting a signal using the LFA. Overall, the LFA is limited in multiplexing ability, is prone to the Hook effect, and lacks the ability for sensitive detection when using low sample volumes [39,40].

The above limitations can potentially be overcome by using a vertical flow immunoassay (VFA), which is comprised of stacked nitrocellulose membranes (NCM) and absorbent pads to create a system for the sample to pass through vertically, thus allowing rapid, high multiplexing capability, high sensitivity, and larger volume capacity [41]. VFA sensors are also highly portable and require minimal infrastructure to engineer and use in health care settings. Moreover, VFAs do not require substrate or conjugate pads, which reduces their cost in comparison to the LFA, which requires both pad types [40]. In this work, we developed the proof of concept for a rapid IL-6 VFA with a wide reportable range that supports its use in a broad spectrum of diseases, including RA, COVID-19-induced CRS, CAR-T therapy-induced CRS, sepsis disease activity, and other inflammatory diseases. The ability to assay IL-6 at the point of care (or even at home) in an ultra-sensitive manner that previous technologies could not achieve can potentially translate to a significant reduction in mortality and morbidity across a wide range of life-threatening inflammatory diseases.

## 2. Materials and Methods

### 2.1. Materials

The rapid vertical flow immunoassay’s plastic housing was purchased from Cytodiagnostics. The following seven different makes of NCMs were purchased for screening: Cytiva (Amersham 0.45 µm, 0.2 µm, and 0.1 µm pore sizes), BioRad (0.45 µm pore size), and MDI (0.8 µm, 0.45 µm, and 0.3 µm pore sizes). Absorbent pads were purchased from Cytiva (Emeryville, CA, USA) (CF7 and CF5). Polyethylene glycol (PEG 3350), polyvinylpyrrolidone (PVP40), bovine serum albumin (BSA), Tris-HCl, and phosphate buffer saline (PBS, pH 7.4), Ethylenediaminetetraacetic acid (EDTA), gelatin, and sodium chloride (NaCl), glucose, and glycine were purchased from Sigma Aldrich (St. Louis, MO, USA). Tween-20 was purchased from Promega Corporation (Madison, WI, USA). Creatinine, recombinant interleukin-8, interferon-gamma inducible protein-10 (IP-10), and TNF-*α* were purchased from R&D Systems (Minneapolis, MN, USA) for specificity evaluation. The anti-IL-6 capture antibody, recombinant IL-6 protein, biotinylated anti-IL-6 detection antibody, and the human IL-6 DuoSet ELISA kit were purchased from R&D Systems. In addition, 150 nm (Lot# HKE0591) and 40 nm (Lot BJB0015) streptavidin-conjugated GNP were purchased from Nanocomposix (San Diego, CA, USA). The same lot number was used to complete the research to avoid any lot-to-lot variability.

### 2.2. Patients and Biosamples

In total, 20 adult healthy control serum samples were obtained from the UT Southwestern Medical Center with informed consent (Dallas, TX, USA). In total, 20 adult RA serum samples were purchased from BioIVT (New York, NY, USA). Serum samples were aliquoted and stored at −20 °C. Before use, thawed serum samples were centrifuged at 5000 rpm for 5 min to remove any lipids or debris, and only the clear serum was applied to the assay. The demographics associated with the 40 samples are summarized in Table 1.

### 2.3. Assay Protocol-I: rIL-6 Detection in Ten-Fold Diluted Healthy Serum

For the detection of recombinant IL-6 (rIL-6) in serum, pooled healthy serum was first ten-fold diluted in serum diluent (9 mM Tris-HCl, 67.5 mM NaCl, 1.5 mM EDTA, 1.5% BSA, 0.12% Tween-20, 0.05% gelatin, 0.25% PEG, pH 8, filtered), and then spiked with rIL-6; 100 pg/mL was used as a positive control; the non-spiked serum was used as a negative control. It was found that using an absorbent pad of size 2.2 cm × 2.7 cm and a membrane of size 1.5 cm × 1.5 cm yielded a clear signal with minimal GNP aggregation. VFAs were first assembled by clamping two layers of NCM at the top, one layer of CF7, and two layers of CF5 at the bottom within the plastic housing. The following protocol was adapted for preliminary screening of membrane pore size. In total, 1 µL of 1 mg/mL of anti-IL-6 capture antibody in PBS was manually spotted at the top left quadrant of the NCM. The NCM was air-blown for one hour. After that, the cartridge was stored at room temperature with desiccator pouches until needed. To run the VFA, the NCM was washed using 100 µL of the washing buffer (20 mM Tris-HCl, 150 mM NaCl, 1% BSA, 0.1% Tween-20, pH 8, filtered). Then, 200 µL of blocking reagent (3 mM EDTA, 3% BSA, 0.1% gelatin, filtered) was applied to the NCM for 5 min. Following an additional 100 µL wash, 100 µL of 10-fold diluted serum sample (positive control: final concentration 100 pg/mL) as prepared above was loaded to the top left quadrant of the NCM. Following an additional 100 µL wash, 10 µL of 10 µg/mL biotinylated anti-IL-6 detection antibody in GNP diluent (20 mM Tris-HCl, 150 mM NaCl, 0.5% PVP40, 1% BSA, 0.19% Tween-20, pH 8, filtered) was loaded into the top left quadrant of the NCM for 30 s. Following an additional 100 µL of wash, 10 µL of 40 nm streptavidin-conjugated GNP (SA-GNP) (OD = 5) in GNP diluent was loaded into the top left quadrant. After 30 s, 400 µL washing buffer was applied to remove any background.

### 2.4. Assay Protocol-II: rIL-6 Detection in Spiked Buffer and Spiked Pooled Healthy Serum for the Evaluation of Detection Limit

For the detection rIL-6 in buffer, rIL-6 was spiked into the serum diluent to constitute 0, 10, 100, 1000, and 10,000 pg/mL. For the detection of rIL-6 in serum, pooled healthy serum was first ten-fold diluted in serum diluent and spiked with recombinant rIL-6 to constitute 0, 1, 3.2, 10, 32, 50, 100, 200, 320, 400, 600, 800, 1000, 3200, and 10,000 pg/mL. The VFAs were first assembled by clamping two layers of Amersham 0.2 µm NCM at the top, one layer of CF7, and two layers of CF5 at the bottom within the plastic housing. The assay protocol follows Figure 1, and the detailed steps are as follows. First, 1 µL of 1 mg/mL anti-IL-6 capture antibody in PBS was spotted at the top left quadrant of the NCM. Next, the NCM was air-blown for one hour. After that, the cartridge was stored at room temperature with desiccator pouches until needed. To run the VFA, the NCM was washed as described above. Then, 200 µL of blocking reagent was applied to the NCM for 5 min. Following an additional wash, 500 µL of samples (buffer or ten-fold diluted serum) as prepared above were loaded onto the entire NCM. After absorbing the excess sample, 100 µL washing buffer was applied, followed by 20 µL 10 µg/mL of biotinylated anti-IL-6 detection antibody in GNP diluent, loaded into the top left quadrant of the NCM for 30 s. After an additional wash, 20 µL of 40 nm SA-GNP (OD = 5) in GNP diluent was loaded. After 30 s, 400 µL washing buffer was applied to remove any background.

### 2.5. Assay Protocol-III: IL-6 Detection in Patient Samples and Healthy Serum

To assay IL-6 level in RA patients and HC serum samples, 1 µL of 1 mg/mL anti-IL-6 capture antibody in PBS was spotted at the top left quadrant, referred to as the test zone; 1 µL of 0.5 mg/mL biotinylated-BSA in PBS, 5% ethanol was spotted at the top right quadrant, referred as Std-1; 1 µL of 0.1 mg/mL biotinylated-BSA in PBS, 5% ethanol was spotted at the lower left quadrant referred as Std-2; 1 µL of PBS was spotted at the lower right quadrant, referred as Std-3. Protocol-III is similar to Protocol-II except for the following changes. After loading 20 µL SA-GNP in GNP diluent to the top left quadrant for 30 s, 200 µL of washing buffer was applied. Then, 0.5 µL of 40 nm SA-GNP (OD = 5) in GNP diluent was added to the top right quadrant, 0.5 µL to the lower left quadrant, and 0.5 µL to the lower right quadrant, followed by 200 µL of washing buffer was loaded to remove any background. The semiquantitative observer score (OS) of IL-6 levels in clinical samples was assigned using the following criteria: if the signal intensity at the test zone is equal to that of Std-3, it is recorded as ‘0’; if higher than Std-3 but lower than Std-2, OS is recorded as ‘+1’; if intensity is equal to Std-2, OS is recorded as ‘+2’; if intensity is higher than that of Std-2 but lower than Std-1, OS is recorded as ‘+3’; if intensity is equal to Std-1, OS is recorded as ‘+4’; if intensity is higher than all 3 standards, OS is recorded as ‘+5’, in Figure 2.

The algorithm used for assigning the semi-quantitative scores corresponding to the smartphone-reported imaging score (IS) was identical to the algorithm described above for the OS, with one exception. For each absolute value captured by the phone at the test zone (TZ), 15% coefficient of variation was allowed. Thus, for example, the IS is reported as “+1” if 0.85 × TZ and 1.15 × TZ were both higher than Std-3 intensity but lower than Std-2; it is recorded as “+2” if 0.85 × TZ was lower than Std-2 but 1.15 × TZ was greater than Std-2; recorded as “+3” if 0.85 × TZ and 1.15 × TZ were both higher than Std-2 intensity but lower than Std-1 intensity. Similar reporting was used across all included standards, not unlike the algorithm used for interpreting the OS.

### 2.6. Comparison of the IL-6 VFA with ELISA

The engineered IL-6 VFA was compared with the conventional ELISA. A commercial ELISA kit (R&D Systems) was used to screen 20 RA and 20 HC serum samples. The serum samples were diluted 10 folds and assayed based on kit instruction and the plate was then transferred to a microplate reader (ELX808, BioTek Instruments, Winooski, VT, USA) and measured at 450 nm optical density.

### 2.7. Data Analysis

Images of the cartridges were captured using the iScanner app on iPhone 12. The images were first inspected by naked eye and then analyzed using ImageJ by placing a line width of 8 pixels across the center of the test spot where the scale is 1 pixel/unit ratio. The dose-response curves and the biomarker data from VFA and commercial ELISA were plotted and analyzed using GraphPad Prism 5 (GraphPad, San Diego, CA, USA). Biomarker group comparisons of VFA and commercial ELISA were analyzed using the Mann–Whitney U-test as datasets were not normally distributed. The one-way ANOVA non-parametric test was used to analyze the LoD and linearity (R^2^) metrics derived for VFA stability evaluation. The LoD was the lowest concentration that could be detected, represented by the sum of the mean of the blanks (*n* = 2) plus three times the standard deviation of the blanks. Receiver operating characteristic (ROC) curves were plotted and the area under the curve (AUC) was used as a measure of the discriminative power of the assay.

## 3. Results

### 3.1. Evaluation of the Porous Membrane and Number of Layers

A sandwich immunoassay was adopted for the vertical-flow paper-based assay. In this assay, monoclonal anti-IL-6 antibodies and biotinylated polyclonal anti-IL-6 antibodies were used for targeting rIL-6. In the initial testing of the vertical flow assay (VFA), the performance of 150 nm streptavidin-GNP (150-SA-GNP) was evaluated using different makes of nitrocellulose membrane (NCM), using different running buffers, and using different stack layers of NCM in the vertical flow, as shown in Appendix A. However, the signal of rIL-6 using 150-SA-GNP was limited to 1 ng/mL, as shown in Appendix A. Next, the performance of 40 nm streptavidin-GNP (40-SA-GNP) was demonstrated to have a higher sensitivity than the 150-SA-GNP, as shown in Appendix A. Hence, 40-SA-GNP was selected for the remainder of the study and is referred to as SA-GNP.

Following assay protocol-I (see Methods), seven types of NCMs were evaluated by comparing the signal-to-noise ratio (SNR) when assaying 10-fold diluted serum spiked with the analyte (100 pg/mL) or non-spiked (0 pg/mL). In this test, triplicates of each type of membrane were evaluated. Figure 3A shows the quantification of SNR using seven types of NCM, while Figure 3B summarizes the SNRs using the different types of NCM. The Amersham membrane, with a pore size of 0.2 µm, was selected as the optimal NCM based on its higher SNR and better morphology of spots. After selecting the make of the NCM for this assay, the number of layers of Amersham 0.2 µm to be used in the vertical flow stack was evaluated. As mentioned above, one, two, and three layers of Amersham 0.2 µm were assembled with one layer of CF7 and two layers of CF5 individually. The NCMs were spotted with capture antibody and tested using spiked and non-spiked serum in triplicates. Our studies show that the two-layer formulation is optimal for rIL-6 detection (please refer to NCMs with a red asterisk in Figure 3A).

### 3.2. Serial Dilution of rIL-6 Detection in Spiked Serum

After optimizing the formulation structure of the VFA and reagent concentrations, the limit of detection (LoD) of rIL-6 in spiked serum was assessed. Following assay protocol-II, ten-fold diluted healthy control serum was spiked with serial dilutions of rIL-6. As demonstrated in Figure 4A, higher concentrations of rIL-6 produced a brighter red color in the top left quadrant. The color intensity of each image was analyzed using ImageJ, and the LoD was identified as the concentration greater than the mean intensity of the negative samples plus three standard deviations above the mean. The dose-response curve showed a wide reportable range of rIL-6 ranging from 10 to 10,000 pg/mL and a LoD of 3.2 pg/mL in serum (Figure 4B). Each point concentration was run in duplicate.

### 3.3. Testing the Reproducibility of the rIL-6 VFA

Following assay protocol-II, the inter-operator coefficient of variance (CV) was examined using assays performed by three researchers. The intra-operator CV was examined using assays performed by one researcher in all cases by replicating standard curves for rIL-6 spiked into the buffer, as shown in Figure 5A. One researcher first ran three serial dilutions of rIL-6 on the VFA (Figure 5B), and the intra-operator CV was calculated, as shown in Figure 5D. Next, two other researchers ran two additional serial dilutions of rIL-6 on the VFA (Figure 5C), and the inter-operator CV was calculated, as shown in Figure 5E. The relatively low inter- and intra-operator CV values indicate that the assay results are reproducible across multiple researchers and when repeatedly performed by a single researcher.

### 3.4. Verifying the Stability of the rIL-6 VFA

To examine the impact of room temperature storage stability on VFA performance, VFA assay kits generated as described above were stored in desiccator bags at room temperature, marked as one-day, two-week, four-week, and six-week, referring to the duration they were kept at room temperature. A serial dilution of rIL-6 was tested on the stored VFA test kits, stored for varying lengths of time as indicated. The pictures in Figure 6A and the quantification data in Figure 6B demonstrate the impact of storage at room temperature. All four linear regression plots showed the same LoD (10 pg/mL) and similar linearity, indicating that the IL-6 VFA assay maintains its performance characteristics for at least six weeks at room temperature.

### 3.5. Specificity of rIL-6 VFA

Glucose (1 mg/mL), glycine (30 mg/mL) and creatinine (13.5 µg/mL) were used to evaluate the effect of interferents on rIL-6 detection, while interleukin-8 (IL-8; 200 pg/mL), interferon-gamma inducible protein-10 (IP-10; 200 pg/mL) and TNF-α (200 pg/mL) were used to evaluate the selectivity of the rIL-6 VFA. We found that the addition of glucose, creatinine, and glycine to the rIL-6 -piked buffer (50 pg/mL) had no significant changes in the intensity of the test zone measurement compared to the rIL-6-spiked buffer. Furthermore, IP-10, IL-8, and TNF-α spiked buffer showed significantly lower intensities than the rIL-6-spiked buffer, and these intensities were similar to the intensities elicited by the non-spiked buffer. These results, as shown in Figure 7, indicate that the VFA is specific for the IL-6 detection. All compounds were tested in triplicates.

### 3.6. Using the Manufactured IL-6 VFA to Assay Human Serum Samples

Given that there is a wide range of IL-6 levels in different inflammatory diseases, additional reference zones of standards (Std-1, Std-2, and Std-3) were introduced into the VFAs to facilitate semi-quantitative readouts, as shown in Figure 2. Std-3 had no biotinylated-BSA (“blank;” bottom right quadrant), while Std-2 (bottom left quadrant) and Std-3 (top right quadrant) had increasing concentrations of biotinylated-BSA spotted at these sites to serve as concentration standards. Thus, when SA-GNP is added, Std-3, Std-2, and Std-1 will show increasing spot intensities. The IL-6 concentration of Std-3 indicates blank, that of Std-2 is comparable to the signal from 10 pg/mL of IL-6, and that of Std-1 is comparable to the signal from 100 pg/mL of IL-6. We tested 20 RA and 20 HC serum using the IL-6 VFA, following assay protocol III (see Methods). As shown in Figure 8A,B, all VFAs showed varying signals on the TZ (top left quadrant), a strong signal for Std-1 (top right quadrant), an intermediate signal for Std-2 (bottom left quadrant), and no signal on Std-3 (bottom right quadrant). Each subject’s IL-6 level after ten-fold dilution was accorded a semi-quantitative observer score (OS), ranging over 0, 1+, 2+, 3+, 4+, and 5+ as detailed in Methods, by three observers, and the imaging scores (IS) of the same 40 samples were recorded (following the method outlined in Figure 1), in Figure 8C. Based on the OS, 12 out of 20 RA patients exhibited 2+ or stronger IL-6 readings, while 2 of the 20 healthy subjects exhibited 2+ or stronger IL-6 readings, indicating that RA patients tended to have higher serum IL-6 levels, with the difference attaining significance (Mann Whitney test *p*-value = 0.0043).

### 3.7. Comparing the Diagnostic Potential of the IL-6 VFA and ELISA

The same serum samples from 20 RA patients and 20 HC subjects tested using the IL-6 VFA were also evaluated using a commercial ELISA for IL-6. The IL-6 levels in RA patients showed a significantly higher value than HC (** *p* < 0.01, Figure 9A), as did the imaging scores (IS) and observer scores (OS) for the IL-6 VFA (** *p* < 0.01, Figure 9B,C). As shown in Figure 9D, a strong correlation between the IS and OS IL-6 scores was observed (a Pearson correlation of 0.88). The correlation between the IL-6 levels assayed by ELISA and VFA was modest (a Spearman correlation of 0.56), as displayed in Figure 9E.

## 4. Discussion

Although vertical flow assays (VFAs) have been reported in the past, this is the first demonstration incorporating three control or internal standard zones to achieve ultra-sensitive semi-quantitative detection of the cytokine IL-6. Several parameters were tested to optimize the vertical flow scheme adopted, including the membrane type (pore size and vendors), number of membranes, GNP reporter size, and the serum dilution factor. The best membrane to use was determined to be Cytiva Amersham with a pore size of 0.2 µm. The absorbent pads used were Cytiva CF7 and CF5. The use of two layers of NCM, two layers of the CF7, and one layer of the CF5 enabled the background noise to be extremely low, reduced GNP aggregation, and facilitated a higher volume capacity.

As seen in the results obtained, IL-6 can be detected in serum using a VFA within 15–20 min. The 5 min difference in this range occurred due to differences in the time taken for the multiple washing and sample loading steps among the different operators. Overall, it was determined that waiting 30 s after loading biotinylated anti-IL-6 antibody to NCM and waiting 30 s after loading SA-GNP to NCM are both crucial to maintaining a low background signal. Under these conditions, the median assay time was 18 min. The LoD of rIL-6 in spiked serum was determined to be 3.2 pg/mL, with a wide reportable range of up to 10,000 pg/mL of rIL-6. The optimization of the membrane pore size and flow rate allows this rapid detection to be possible. A higher flow speed and smaller membrane pore size are proven to aid in increasing the sensitivity of the system, allowing the detection of biomarkers in a faster and more efficient manner.

Although ELISA is a widely used diagnostic test in laboratories, it has multiple limitations. ELISA is time-consuming, taking hours to days to complete, usually requires a plate-reader, and demands professionals to operate. The VFA, on the other hand, provides rapid results, requires fewer steps than ELISA, making it much more user-friendly, and is less expensive, making it particularly helpful in low-resource settings. Additionally, compared to the LFA, the VFA exhibits higher multiplexing capabilities, elimination of the hook effect (false negatives), higher sample capacity, and a relatively higher detection speed. These render the VFA a promising platform to explore for future diagnostics.

As seen in the tests conducted to assess the reproducibility and stability of the VFA, the inter- and intra-operator CV do not vary significantly, indicating that the VFA results can be easily reproduced. In addition, the storage condition had no significant impact on assay performance for at least six weeks. This indicates that the results obtained are reliable and that storing the VFA does not significantly impact its accuracy or reliability. The engineered VFA also performs well with clinical samples, as shown by the OS readouts of VFA being in good agreement with the IS and ELISA readouts. These factors render the VFA an appropriate platform for detecting IL-6 in clinical samples within a much shorter period than ELISA and with higher sensitivity than the LFA.

The study has potential limitations. With a small sample size, it is a possibility that our results may have overestimated the diagnostic capability of the engineered IL-6 VFA. Therefore, more comprehensive clinical validation with larger sample sizes is warranted. Additionally, the scope of the study can be expanded by screening for other inflammatory diseases, including COVID-19-induced CRS, sepsis, and cancers. More importantly, a head-to-head comparison of IL-6 VFA to LFA to ELISA using the same batch of clinical samples would serve as a direct demonstration of the diagnostic power of the engineered IL-6 VFA. Moreover, not investigated are modifications to the assay protocol so that blood can be used directly with these tests at the POC. In this study, we have used cleared serum as opposed to un-cleared samples to potentially prevent any lipids or cell debris from interfering with analyte detection or clogging the small-sized pores of the membranes in the VFA. At the POC testing, available options include the use of reagents such as Cleanascite, a reagent that selectively removes lipids, fats, impurities, and other cell debris from samples. Studies are warranted to assess if such procedures are indeed necessary for the VFA to work optimally. In this study, the capture antibody was manually spotted onto the membrane; however, an automated dispensing machine that could quantitatively dispense the capture antibody in a precise volume and constant location would be preferred. This can vastly reduce spot-to-spot variation in terms of area and shape and eliminate any potential membrane trauma caused by manual spotting.

## 5. Conclusions

Through the results obtained from this study and the data analyzed, it can be concluded that VFA could be a valuable tool for detecting IL-6 in serum. The VFA platform is quick, efficient, and on par with ELISA in terms of accuracy and sensitivity, particularly given that ELISA is a procedure that takes hours to days to complete. Furthermore, the VFA has a low cost, a fast response time, and is extremely easy to use. This makes it advantageous over other serum IL-6 detection approaches, as well as for assaying other blood cytokines. The LoD of rIL-6 in spiked serum was determined to be 3.2 pg/mL, with a wide reportable range of up to 10,000 pg/mL of rIL-6. We have thus demonstrated that the engineered VFA is able to detect IL-6 in most of the inflammatory diseases, allowing early detection of a rise in IL-6. Detecting an early rise in IL-6 levels could be of clinical use in triaging patients for admission to hospital/ICU, for commencing IL-6 targeted therapeutics, or for assessing disease activity in multiple life-threatening diseases.

## 6. Patents

There is a patent application resulting from the work reported in this manuscript. Patent owner 1: Chandra Mohan and owner 2: Rongwei Lei.

## Figures and Tables

**Figure 1 biosensors-12-00756-f001:**
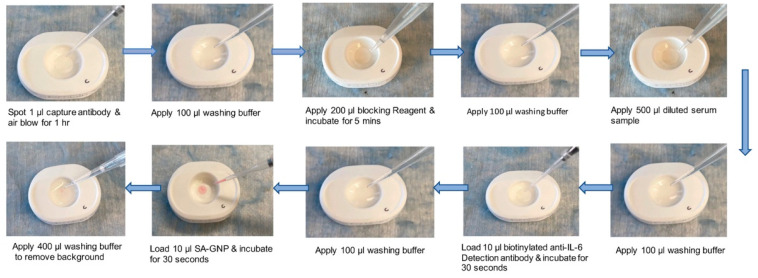
Vertical-flow paper-based assays for IL-6 detection. Depicted is the workflow for assaying human serum IL-6 using the vertical-flow paper-based assay. The paper-based assays are prepared by stacking two layers of 0.2 µm nitrocellulose membrane (NCM), one layer of CF7, and two layers of CF5. This combination of NCM and absorbent pads is clamped together to secure them into the plastic housing. The workflow starts with a wash, followed by adding blocking buffer, loading the sample, adding detection antibody, and then adding the GNP. There is a wash step in between each of the addition steps. The signal is observed and quantified after the final wash.

**Figure 2 biosensors-12-00756-f002:**
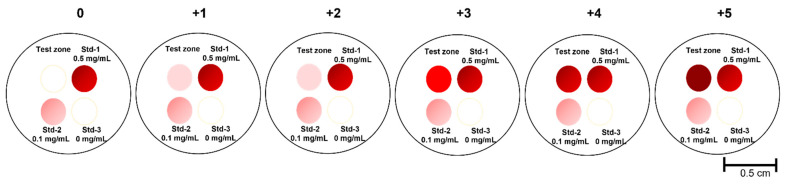
Semi-quantitative determination of IL-6 using three reference standards. Shown is a schematic of the IL-6 VFA demonstrating a test zone (top left) and three reference zones loaded with different concentrations of IL-6 standard. The observer score (OS) from 0 to +5 are indicated at the top of each image to show how this semi-quantitative score was assigned based on the relative intensities seen at the test zone and three reference zones included.

**Figure 3 biosensors-12-00756-f003:**
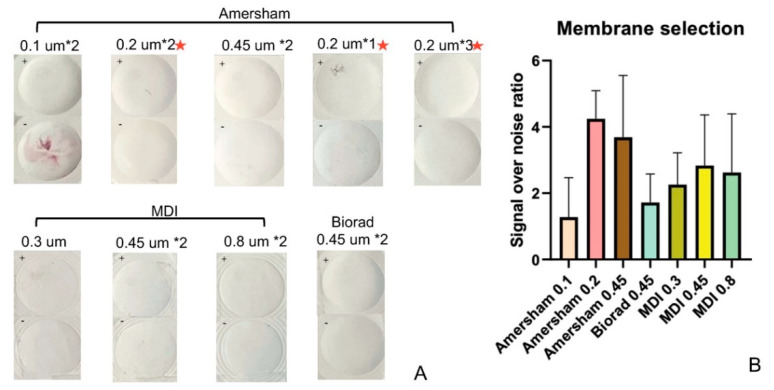
Effect of NCM membrane pore sizes and layers on assay performance. (**A**) The two rows depict the seven types of porous membranes that were evaluated to detect 100 pg/mL rIL-6 in 10-fold-diluted serum and non-spiked 10-fold diluted serum control. Triplicates were analyzed. Each label within the figure displays the manufacturer from which the membrane was sourced, the pore size, and the number of layers of membrane tested. The test zone is at the top left corner marked as + (spiked with 100 pg/mL rIL-6) or as—(non-spiked sample). The + (100 pg/mL) test zone should appear as a homogenous, uniform pink spot with maximum intensity, while the—(0 pg/mL) test zone should show no signal. It is clear that the Amersham 0.2 µm membrane displayed the best intensity for the + control (100 pg/mL) and the least background for the—control (0 pg/mL). It was selected and tested further using multiple stacked layers of that membrane, as shown by the middle picture in the first row and the last two pictures in the third row (all marked by red asterisks), which depict the single, double, and triple-stacked layers of Amersham 0.2 µm membrane. The optimal membrane formulation comprised two layers of the Amersham 0.2 µm (Amersham 0.2 µm × 2) since it displayed the strongest, homogenous signal in the positive test zone. (**B**) Colorimetric intensity values were calculated using ImageJ, and the SNR was computed for each type of membrane. Amersham 0.45 µm and 0.2 µm show relatively higher SNR; however, the error bar shows that the Amersham 0.2 µm membrane exhibits less variability. The error bar indicates the standard deviation of three measurements of SNR. SNR: signal-over-noise ratio.

**Figure 4 biosensors-12-00756-f004:**
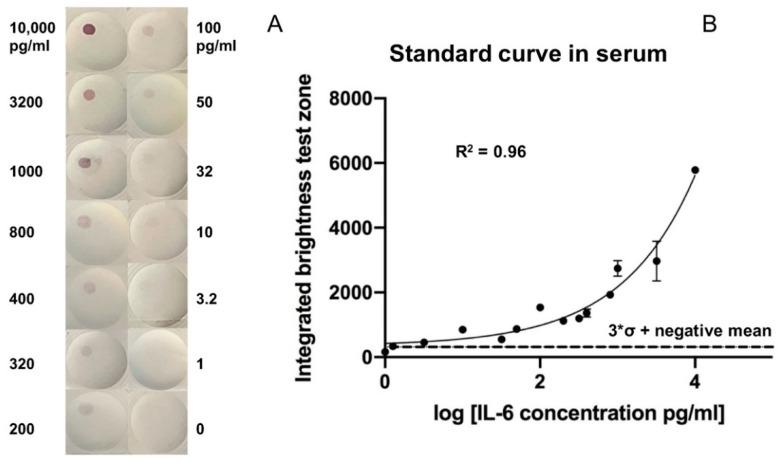
Dose-response curve for detecting rIL-6 spiked into healthy human serum using the vertical-flow paper-based assay. (**A**) The membrane-type used is Amersham 0.2 µm with two layers. Two replicates were obtained at each concentration. The top left of the membrane represents the test zone. (**B**) Colorimetric intensity values were calculated using ImageJ. The 3 × σ + negative mean line indicates the summation of the mean intensity of non-spiked healthy serum samples (0 pg/mL) and 3X its standard deviation (σ). The concentration above this line marks the LoD, which is 3.2 pg/mL. The dose-response curve demonstrated a reportable range from 3.2 pg/mL to 10,000 pg/mL of rIL-6 in ten-fold diluted serum.

**Figure 5 biosensors-12-00756-f005:**
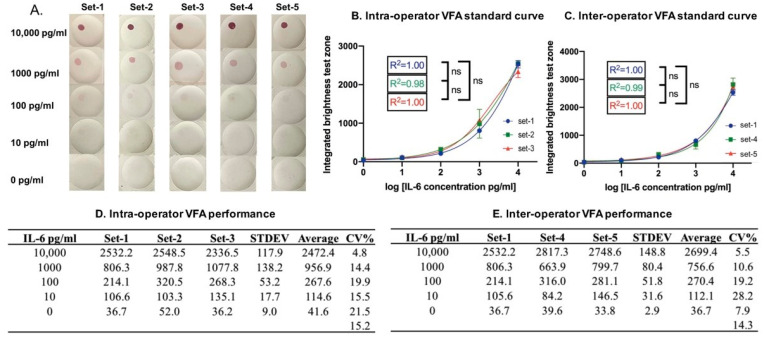
The reproducibility of the Il-6 VFA. (**A**) Images of rIL-6 serial dilutions obtained using the IL-6 VFA, with assays performed by three researchers. IL-6 VFA sets 1, 2, and 3 were performed by the same researcher so that the intra-operator CV for the standard curves can be computed. IL-6 VFA sets 1, 4, and 5 were performed by three different researchers, one per set, to determine the inter-operator CV. The membrane-type used in (**A**) is Amersham 0.2 µm with two layers. (**B**,**C**) Shown are the corresponding dose-response curves for computing intra-operator reproducibility and inter-operator reproducibility. (**D**,**E**) The tables associated with each curve show the inter-operator and intra-operator CV at each point concentration and the average overall CV for the IL-6 VFA assay. The numbers listed within the tables represent the respective imaging score values. The relatively low inter- and intra-operator CV values indicate that the assay results are reproducible across multiple researchers and when repeatedly performed by a single researcher.

**Figure 6 biosensors-12-00756-f006:**
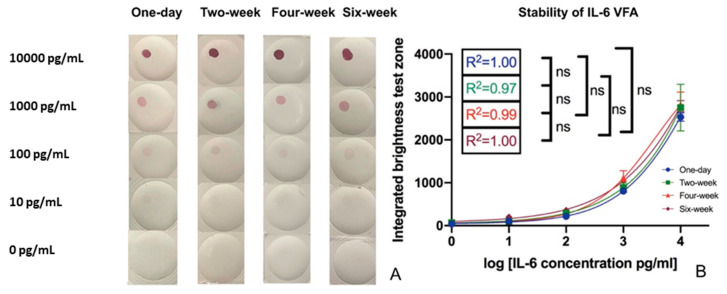
The impact of storage on the performance of the IL-6 VFA. (**A**) IL-6 VFA tests stored at room temperature for one day, two weeks, four weeks, and six weeks were tested to detect varying doses of rIL-6 spiked into buffer using VFAs with Amersham 0.2 µm membrane in two layers. The correlation of rIL-6 levels to the imaging intensity (IS score) of the VFA tests under different storage durations is plotted in (**B**).

**Figure 7 biosensors-12-00756-f007:**
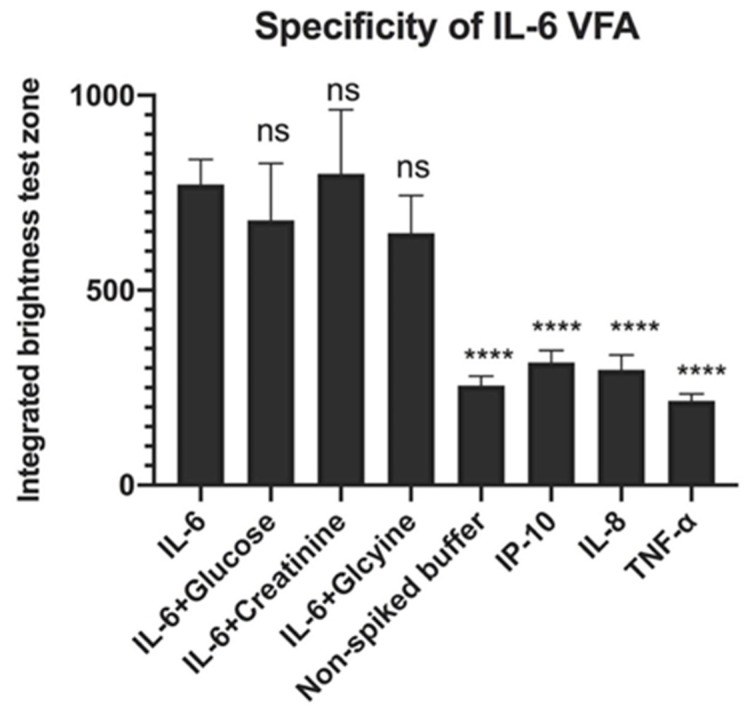
Specificity of the rIL-6 VFA. Addition of glucose, creatinine, and glycine to rIL-6-spiked buffer did not significantly affect the intensity of the signal at the test zone compared to rIL-6-spiked buffer. IP-10, Il-8, and TNF-α spiked buffer showed significantly lower intensities than rIL-6-spiked buffer on the IL-6 VFA and similar intensities to the non-spiked buffer, indicating that the-VFA is relatively specific for IL-6 detection. Samples were loaded onto the VFA in triplicates. ^ns^
*p* > 0.05, **** *p* < 0.0001 compared to the positive control, as determined using one-way ANOVA.

**Figure 8 biosensors-12-00756-f008:**
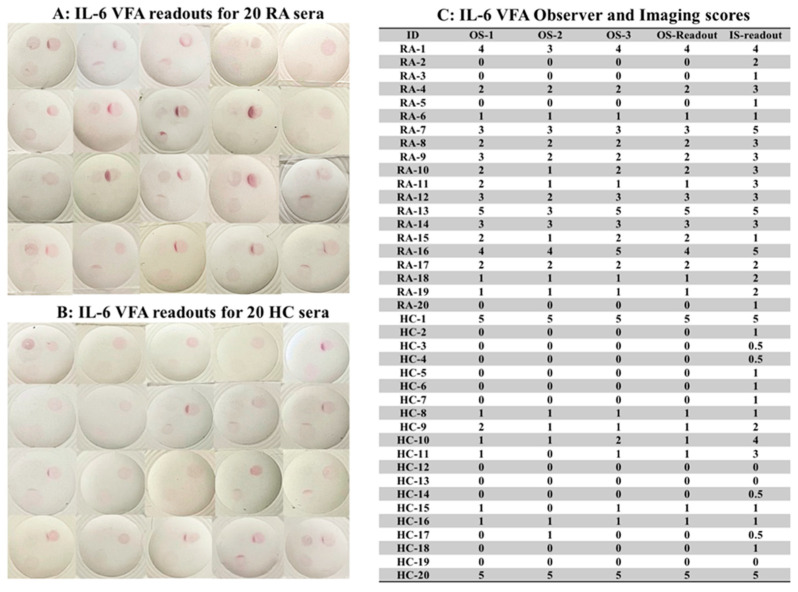
Use of the engineered IL-6 VFA to assay IL-6 in clinical samples. (**A**,**B**) This study tested 20 RA and 20 HC serum samples using the IL-6 VFA, following assay protocol III. Shown are images of the IL-6 VFA run using 20 RA and 20 HC. The test zone is marked at the top left quadrant, Std-1 is included at the top right quadrant, Std-2 is included at the lower left quadrant, and Std-3 is included at the lower right quadrant of the membrane. All VFAs employed Amersham 0.2 µm with two layers. The VFAs showed varying signals at the test zone (top left quadrant), a strong signal at Std-1 (top right quadrant), an intermediate signal at Std-2 (bottom left quadrant), and no signal at Std-3 (bottom right quadrant). (**C**) Each subject’s IL-6 level was accorded a semi-quantitative observer score (OS), ranging from 0, 1+, 2+, 3+, 4+, and 5+; a semi-quantitative imaging score, ranging from 0, 0.5+, 1+, 2+, 3+, 4+, 5+ as detailed in Methods and Figure 1. The OS readout was based on the mean values from two observers. RA: rheumatoid arthritis; HC: healthy control; OS: observer score; IS: imaging score.

**Figure 9 biosensors-12-00756-f009:**
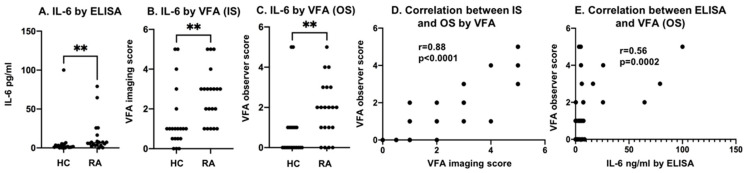
Discriminating power of IL-6 VFA and commercial ELISA. (**A**–**C**) IL-6 levels in 20 RA serum and 20 HC serum as assayed using a commercial IL-6 ELISA, IL-6 VFA with IS readouts, and IL-6 VFA with OS readouts. IL-6 level group comparisons were analyzed using the Mann–Whitney U-test (** *p* < 0.01). IL-6 levels in RA were significantly higher than in HC by all 3 readouts. (**D**) The correlation between VFA by IS and VFA by OS showed a Spearman correlation coefficient of 0.88, *p* < 0.0001. (**E**) The correlation between IL-6 by ELISA and VFA by OS is plotted with a Spearman correlation coefficient = 0.56. *p* = 0.0002. RA: rheumatoid arthritis; HC: healthy control IS: imaging score; OS: observer score.

**Table 1 biosensors-12-00756-t001:** Demographic data of the samples.

	Variable	Healthy Controls (*n* = 20)	Rheumatoid Arthritis (*n* = 20)
Sex	Female	9	6
	Male	11	14
Race	Caucasian	10	17
	African American	6	1
	Other	4	2

## Data Availability

The data presented in this study are available on request from the corresponding author. All data relevant to the study are included in the article or uploaded as supplementary information. The data are not publicly available due to privacy laws. Data are available from the Ethics Committee for researchers who meet the criteria for access to confidential data. Please refer to the corresponding author.

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
