# Peer review of "Ultra-Sensitive and Semi-Quantitative Vertical Flow Assay for the Rapid Detection of Interleukin-6 in Inflammatory Diseases"

_biosensors, 2022, doi:10.3390/bios12090756_

Round 1

Reviewer 1 Report

Lei et al have reported their developed vertical flow assay for the detection of Interleukin-6. The ideal is interesting and they showed convincing results, however, it is not clear the main advantage of the VFA compared to LFA in the detection of only one biomarker here. Other comments are attached below.

(1) Please provide a schematic illustrating how the sensor works for real-time detection of IL-6. Also, the schematic of the operation of this VFA sensor should move to the main manuscript.

(2) In Figure 2, the authors should number each sample. It is not clear the differences among these 6 samples on the top-left corner. All of the scale bars were missing.

(3) The authors claimed that each measurement took 15 min. How did the sensor real-time monitor chronic diseases?

(4) How about the selectivity of the sensor among the potential interferents?

(5) The authors claimed their sensors are ultra-sensitive. However, they did not demonstrate the sensitivity of the sensor. Moreover, in the Conclusion, it is not suitable to demonstrate “the sensitivity of the VFA is low enough to …”

Author Response

(1) Please provide a schematic illustrating how the sensor works for real-time detection of IL-6. Also, the schematic of the operation of this VFA sensor should move to the main manuscript.

Authors: We thank the reviewer for the suggestion. Please see revised Figure 1, where this has been addressed.

(2) In Figure 2, the authors should number each sample. It is not clear the differences among these 6 samples on the top-left corner. All of the scale bars were missing.

Authors: We thank the reviewer for the suggestion. Please see revised Figure 2, where this has been addressed.

(3) The authors claimed that each measurement took 15 min. How did the sensor real-time monitor chronic diseases?

Authors: We thank the reviewer for the suggestion. We have used the word “rapid” instead of “real-time” since the assay may require 15 minutes. The word “rapid” is now used in title, abstract, introduction, Methods and Discussion.

(4) How about the selectivity of the sensor among the potential interferents?

Authors: We have now completed additional experiments, please see new Figure 7. Addition of glucose, creatinine, and glycine to rIL-6 spiked buffer did not significantly affect the intensity of the signal at the test zone, compared to rIL-6 spiked buffer without interferent. IP-10, Il-8, and TNF-α spiked buffer showed significantly lower intensities than rIL-6 spiked buffer on the IL-6 VFA, and these intensities were similar to the intensities elicited by the non-spiked buffer, indicating that the VFA is relatively specific for IL-6 detection.

(5) The authors claimed their sensors are ultra-sensitive. However, they did not demonstrate the sensitivity of the sensor. Moreover, in Conclusion, it is not suitable to demonstrate that “the sensitivity of the VFA is low enough to …”

Authors: The LoD of rIL-6 in spiked serum was determined to be 3.2 pg/mL with a wide reportable range up to 10,000 pg/mL of rIL-6. We have thus demonstrated that the engineered VFA is able to detect the IL-6 in the most of the inflammation diseases. These revised sentences have been added to the Conclusion.

Reviewer 2 Report

In the manuscript entitled Ultra-sensitive and semi-quantitative vertical flow assay for the rapid detection of Interleukin-6 in inflammatory diseases although the author does provide an interesting solution to detect IL6. However the overall irregularities and variation of the images in the results needs to be improved. Also, the correlation of the data as positive and negative gives me a light faith in the manuscript.

I feel the experiments need to be done high time for better data acquisition and representation. The overall flow of the manuscript is confusing and does not point to a significant result as the Spearman correlation value between ELISA and VFA is just 0.56.

Some of my major queries for revision of these manuscripts are as follows:

Author mentions the use of 40 nm SA-GNP (OD=5) throughout the manuscript however the interaction between the IL 6 antibody and the GNP is unclear.

Also, the reason for using a GNP with OD=5 needs to be justified as the average OD reported for use in flow-based assay is around 1-2

An additional washing step is added during the performance of VFA which seems to be unnecessary.

The author states in line 161 the use of 5% ethanol for spotting at the top right quadrant, referred as Std-1; However, I do have some serious doubts about use of organic solvents such as ethanol on NCM.

The images of figure 1 for IL-6 VFA demonstrating test zones and three reference zones is inconsistent in overall manuscript. I have rather little faith looking at the consistency of the results.  

Also, the need for two NCM needs to be justified as one NCM is enough to provide the results in vertical flow-based assay.

It is important to discuss why the sensitivity reported for 40-SA-GNP was higher than for 150-SA-GNP.

In supplementary  Fig A2, Why nanoparticles looks blue? It seems aggregated.

In standard curve, the value range is too broad. How will be accurate for quantitative analysis? Authors should perform a close range of concentrations.

Author Response

I feel the experiments need to be done high time for better data acquisition and representation. The overall flow of the manuscript is confusing and does not point to a significant result, as the Spearman correlation value between ELISA and VFA is just 0.56.

Authors: We very much appreciate the Reviewer’s concern and have now inserted a cartoon schematic in Figure 1 to better illustrate the overall flow the report and result interpretation. The Spearman correlation value is 0.56 which may seem low upon initial review. However, the novel VFA was used to determine the concentration of IL-6 using a semi-quantitative method which may be one reason for a modest correlation. Furthermore, Rheumatoid Arthritis is a condition where the patients’ IL-6 concentration is less than 10 fold greater than that in healthy controls. Due to this, the correlation values between any 2 assay platforms may tend to be lower than it might have been if the IL-6 differences between patients and healthy controls were more striking.  Finally, with a modest correlation of 0.56, we do not really know whether ELISA or the VFA is better reflective of the ground truth, without having some other assay to ascertain exact concentration (eg. quantitative mass spectrometry).

1) Author mentions the use of 40 nm SA-GNP (OD=5) throughout the manuscript however, the interaction between the IL 6 antibody and the GNP is unclear.

Authors: The IL-6 antibody is biotinylated and the gold nanoparticles are streptavidin conjugated. Thus, the biotin and streptavidin bind together after the gold nanoparticles are applied to the IL-6 antibody, and together they produce a colorimetric signal. The use of this 2-step detection approach amplifies the signals obtained.

2) Also, the reason for using a GNP with OD=5 needs to be justified as the average OD reported for use in the flow-based assay is around 1-2

Authors: Since the VFA uses a semi-quantitative detection method, researchers rely heavily on the visibility of a signal. While OD values of 1-2 can produce signals, the signal elicited were dim at 100 pg/ml and there was no signal at 10 pg/ml in our preliminary tests. Additionally, the degree of signal differences across different concentrations was insignificant. Therefore, we opted to use an OD value of 5 because it produces an intense signal that visibly varies across different concentration points in a consistent and reproducible  manner.

3) An additional washing step is added during the performance of VFA, which seems to be unnecessary

Authors: The washing step can remove unbound materials from the previous step that are retained on the membrane so as to avoid the potential clogging/aggregation of GNP at the last step. We believe this is particularly important because we are utilizing a small pore size membrane of 0.2 um.

4) The author states in line 161 the use of 5% ethanol for spotting at the top right quadrant, referred to as Std-1; However, I do have some serious doubts about the use of organic solvents such as ethanol on NCM.

Authors: On the nitrocellulose membrane (NCM), 5%-20% ethanol or methanol can promote protein binding to the membrane thus yielding a better signal. Also, others have used sucrose or trehalose supplemented antibody for stability on the membrane. Because our antibody was stored in 1xPBS, 1% trehalose, we just supplemented this with 5% ethanol for faster immobilization. This is a common approach in immobilizing antibodies in lateral flow assay membrane (Rapid Lateral Flow Test Strips - Merck Milliporehttps://www.merckmillipore.com › ru_RU › USD).

5) The images in figure 1 for IL-6 VFA demonstrating test zones and three reference zones is inconsistent in the overall manuscript. I have rather little faith in looking at the consistency of the results.  

Authors: We understand the Reviewer’s concern. The “inconsistency” in the test zone is because the IL-6 levels are different in different test samples. The occasional heterogeneity noted in Std-1 may have been caused by the possible spreading of GNP from the test zone to the Std-1 zone, perhaps due to titling or handshaking. Our planned use of a kinematic automated dispenser (in future studies) to accurately dispense reagent dots may help reduce variance and heterogeneity in the test zones. Nevertheless, this has not compromised the overall performance of this assay in measuring IL-6.

6) Also, the need for two NCMs needs to be justified as one NCM is enough to provide the results in a vertical flow-based assay.

Authors: Although one layer of NCM seems to provide a uniform signal it is really faint and is difficult to visualize, as shown in Figure 2. Two NCM layers were used as they generated the least background, the most homogenous signal, and higher signal over noise ratios. Interestingly, some authors have even used 5 layers of NCM to attain consistent flow, alluding to the important contribution of multiple NCM layers (Kim, S., Hao, Y., Miller, E. A., Tay, D., Yee, E., Kongsuphol, P., Jia, H., McBee, M., Preiser, P. R., & Sikes, H. D. (2021). Vertical Flow Cellulose-Based Assays for SARS-CoV-2 Antibody Detection in Human Serum. ACS sensors, 6(5), 1891–1898. https://doi.org/10.1021/acssensors.1c00235)

7) It is important to discuss why the sensitivity reported for 40-SA-GNP was higher than for 150-SA-GNP.

Authors: We envision that the 40-SA-GNP, being smaller, may readily pass through the small-sized pore NCM (0.2 um) more easily with washing, resulting in faster turnaround time. The signal may also increase with an increase in 40-SA-GNP concentration without increasing the background. Due to this, they may provide better sensitivity than the 150-SA-GNP. However, a more systematic study would be needed to verify this hypothesis and to elucidate other possible reason(s) for the observed differences.

8) In supplementary  Fig A2, Why do nanoparticles look blue? It seems aggregated.

Authors: The 150-SA-GNP nanoparticles (OD=10) should exhibit a uniform green/blue signal upon dispensing on the membrane. The nanoparticles appear blue because after they are diluted in buffer (OD<10), their color intensity weakens towards a blue-green color.

9) In the standard curve, the value range is too broad. How will be accurate for quantitative analysis? Authors should perform a close range of concentrations.

Authors: Thank you for your suggestion. We have added more tested concentrations in the standard curve, shown in the Figure 4.  In Table A1 in the Appendix, we list the wide range of serum IL-6 concentrations in different inflammatory diseases, ranging from 10-10,000 pg/ml. For example, in sepsis, serum IL-6 could surge from 100 pg/ml to 1000 pg/ml within the first 5 hours. In several other diseases, they are typically less than 100 pg/ml. The updated standard curve shown in Figure 4 covers these potential IL-6 concentrations.

Reviewer 3 Report

Manuscript ID: biosensors-1845402

Title:  Ultra-sensitive and semi-quantitative vertical flow assay for the rapid detection of Interleukin-6 in inflammatory diseases

General comments: In this manuscript, the authors report a vertical flow immunoassay system for the detection of inflammatory biomarker IL-6. Although the optimization of gold nanoparticles size, and membrane pore sizes have been executed and data have been presented to support the experiments, the novelty of this work is not sufficient, and several problems must be explained and tackled. there are some major issues that need to be addressed before further consideration. Therefore, major revision or rejection is recommended.

1)    The innovation of the manuscript should be improved in the introduction section.

2)    Author reported that IL-6 by ELISA requires two days to complete. There are a lot of commercially available antibody-coated ELISA kits available to perform the IL-6 assay within 90 mins to 3 hours. I suggest the author should revise the respective information in the introduction.

3)    In the introduction section, author should focus on the previous diagnostic approaches instead of explaining the inflammatory biomarker of IL-6. There are a number of studies such as colorimetric, and fluorometric labels that have been used to detect IL-6 and they obtained the LOD as low as 5 pg/mL with serum samples. I suggest to the author talk about the previous technologies and what are the advantages of your diagnostic platform compared with the previous technologies.

4)    Did the author find any issues in using the thawed serum samples? Why the clear serum required to perform the assay? Author claimed that this device is POC testing. If so, how the lipids or debris will be removed in POC approach?

5)    In this study biotinylated Anti-IL-6 detection antibody and streptavidin-conjugated GNP has been used to detect the IL-6. Using single detection Ab could eliminate a few steps. Why did author not use GNP conjugated Anti-IL-6 Ab as a detection antibody?

6)    Are there any differences in the device assembly or membrane pore size between Assay I and Assay II? No need to repeat the device assembly steps for each assay method, I suggest author to make a section for device assembly and membrane preparation instead of repeating it in each assay.

7)    Why author chose 10- fold diluted serum to spiked with the analyte?

8)    One replicates of membrane images for each layering formulation is sufficient for Figure 2 (a) instead of giving three replicates of membrane pictures.  I suggest the author to include the quantitative results of Amersham 0.2 um, single, double and triple layer formulations.

9)    I suggest author keep the single replicate of membrane images in Figures 3 (a),4 (a), and 5 (a) instead of showing two replicates of membrane images. Adding standard deviation in the graph for each concentration will give better information to the readers than the qualitative images.

10) The Inter and Intra CV acceptable range is 10 to 15%, respectively. However, the author got a higher CV% (for 100 and 10 pg/mL) for inter and Intra operator VFA performance. What strategy will be followed to improve the inter and intra-assay CV%?

11) Figure 6 a and b exhibited the heterogeneous color development on the Std-1. Why does the human serum generate a heterogeneous color distribution in the VFA system?

12) It is suggested to give some discussion on the improved detection limit by the VFA system as a comparison to LFI?

13) How are the capture antibodies spotted on the nitrocellulose membrane? how accurate and reproducible are the antibodies spot if the antibodies are spotted on the membrane manually?

14)  Assay I and assay II procedure are the same to perform the assay in the VFA device.  Author should combine these two sections as a single that to improve the flow of the manuscript.

15) How was the signal intensity of the membrane measured? Did author measure the whole spot or any specific pixel for each membrane has measured?

Author Response

In this manuscript, the authors report a vertical flow immunoassay system for the detection of inflammatory biomarker IL-6. Although the optimization of gold nanoparticles size, and membrane pore sizes have been executed and data have been presented to support the experiments, the novelty of this work is not sufficient, and several problems must be explained and tackled. there are some major issues that need to be addressed before further consideration. Therefore, major revision or rejection is recommended.

1)    The innovation of the manuscript should be improved in the introduction section.

Authors: We thank the reviewer for the suggestion. Please refer to the revised introduction section for the implemented changes, in particular the last 3 paragraphs.

2)    Author reported that IL-6 by ELISA requires two days to complete. There are a lot of commercially available antibody-coated ELISA kits available to perform the IL-6 assay within 90 mins to 3 hours. I suggest the author should revise the respective information in the introduction.

Authors: We thank the reviewer for the suggestion. The paragraph on ELISA assays in the Introduction has been revised accordingly.

3)    In the introduction section, author should focus on the previous diagnostic approaches instead of explaining the inflammatory biomarker of IL-6. There are a number of studies such as colorimetric, and fluorometric labels that have been used to detect IL-6 and they obtained the LOD as low as 5 pg/mL with serum samples. I suggest to the author talk about the previous technologies and what are the advantages of your diagnostic platform compared with the previous technologies.

Authors: We thank the reviewer for the suggestion. Please refer to the revised introduction section for the implemented changes, in particular the last 3 paragraphs.

4)    Did the author find any issues in using the thawed serum samples? Why the clear serum required to perform the assay? Author claimed that this device is POC testing. If so, how the lipids or debris will be removed in POC approach?

Authors: We have not observed any issues using thawed serum samples. We have used cleared serum as opposed to un-cleared samples to potentially prevent any lipids or cell debris from interfering with analyte detection or clogging the small-sized pores of the membranes in the VFA. At the POC testing, available options include the use of reagents such as Cleanascite, a reagent that selectively removes lipids, fats, impurities, and other cell debris from samples. Studies are warranted to assess if such procedures are indeed necessary for the VFA to work optimally. This has now been added to the revised Discussion.

5)    In this study biotinylated Anti-IL-6 detection antibody and streptavidin-conjugated GNP has been used to detect the IL-6. Using single detection Ab could eliminate a few steps. Why did author not use GNP conjugated Anti-IL-6 Ab as a detection antibody?

Authors: We did not use a GNP conjugated anti-IL-6 detection antibody because applying the biotinylated detection antibody and the streptavidin-GNP particles separately led to assay amplification. Previous attempts showed that the antibody conjugated GNP could not attain a LoD at 10 pg/ml. Hence, we opted to use the 2-step approach.

6)    Are there any differences in the device assembly or membrane pore size between Assay I and Assay II? No need to repeat the device assembly steps for each assay method, I suggest author to make a section for device assembly and membrane preparation instead of repeating it in each assay.

Authors: There are slight differences in the buffer used and the volume of sample applied. Therefore, we have elected to keep the protocols in both sections to avoid any confusion, and to facilitate full reproducibility of the reported methods.

7)    Why author chose 10- fold diluted serum to spiked with the analyte?

Authors: This serum dilution factor was incorporated for 3 reasons: 1) to avoid any potential membrane clogging, 2) to attain a higher volume of sample in order to cover the whole membrane surface because the test zone spot area is only 1/10 of the window area, and 3) 10-fold diluted serum results in 500 ul volume which is within the capacity of VFA absorption and does not significantly lengthen the assay time.

8)    One replicates of membrane images for each layering formulation is sufficient for Figure 2 (a) instead of giving three replicates of membrane pictures.  I suggest the author to include the quantitative results of Amersham 0.2 um, single, double and triple layer formulations.

Authors: We thank the reviewer for the suggestion. Please see the revised result section (Figure 3) to see the implemented suggestions.

9)    I suggest author keep the single replicate of membrane images in Figures 3 (a),4 (a), and 5 (a) instead of showing two replicates of membrane images. Adding standard deviation in the graph for each concentration will give better information to the readers than the qualitative images.

Authors: We thank the reviewer for the suggestion. Please see the revised result section (Figure 4,5,6) to see the implemented suggestions.

10) The Inter and Intra CV acceptable range is 10 to 15%, respectively. However, the author got a higher CV% (for 100 and 10 pg/mL) for inter and Intra operator VFA performance. What strategy will be followed to improve the inter and intra-assay CV%?

Authors: This is a great question and we have devised a solution. We are in the process of purchasing a kinematic automated dispenser to accurately dispense reagent dots onto the membrane. This can largely reduce the signal area variance, and capture antibody-membrane contact variance. Also, we are arranging for a dry room, which will be a dedicated space for immobilized point of care devices, where the humidity, temperature and light will all be constant. We believe these measures will significantly improve all CV percentages.

11) Figure 6 a and b exhibited the heterogeneous color development on the Std-1. Why does the human serum generate a heterogeneous color distribution in the VFA system?

Authors: The occasional heterogeneity noted in Std-1 may have been caused by the possible spreading of GNP from the test zone to the Std-1 zone, perhaps due to titling or handshaking. The use of a kinematic automated dispenser to accurately dispense reagent dots may also help reduce variance and heterogeneity. Nevertheless, this has not compromised the overall performance of this assay in measuring IL-6.

12) It is suggested to give some discussion on the improved detection limit by the VFA system as a comparison to LFI?

Authors: We thank the reviewer for this suggestion. A previous paper published by the authors describes the improved detection limit of the VFA in comparison to the LFA (23). This discussion has now been added to the background in the revised Introduction explaining disparities between VFA and LFA detection.

13) How are the capture antibodies spotted on the nitrocellulose membrane? how accurate and reproducible are the antibodies spot if the antibodies are spotted on the membrane manually?

Authors: The capture antibodies are spotted onto the membranes manually. Although this may not seem reproducible, the amount of the capture sample is 1 ul at each quadrant, and thus when dispensed, it consistently creates similarly sized circles in all of quadrants of the VFA. If there were any issues with improper capture antibody applied or discrepancies, the VFA was not used for experimentation, but was repeated. Furthermore, we are in the process of purchasing a kinematic automated dispenser to accurately dispense reagent dots onto the membrane. This can largely reduce the signal area variance, and capture antibody-membrane contact variance.  However, the purpose of this paper is to establish the technicality of the VFA and its feasibility. We will further re-assess the diagnostic power and end-user friendliness of the VFA in a follow-up paper once the automated machine becomes available.

14)  Assay I and assay II procedure are the same to perform the assay in the VFA device.  Author should combine these two sections as a single that to improve the flow of the manuscript.

Authors: There are slight differences in the buffer used and the volume of sample applied. Therefore, we have elected to keep the protocols in both sections to avoid any confusion, and to facilitate full reproducibility of the reported methods.

15) How was the signal intensity of the membrane measured? Did author measure the whole spot or any specific pixel for each membrane has measured?

Authors: The intensity of the signals was measured using ImageJ analysis by placing a line width of 8 pixels across the center of the test spot where the scale is 1 pixel/unit ratio. The integrated brightness signal was collected using image J. This has now been added to the revised Methods.

Round 2

Reviewer 1 Report

The authors have addressed the reviewers' comments. I agree to accept this work for publication.

Author Response

Thank you for your review and comments.

Reviewer 2 Report

A complete spell check needs to be done

Also, there is inconsistency in figure number, kindly update it.

Manuscript needs to be checked for grammatical errors.

In introduction section author could cite addition reference showing advantage of vertical flow assay over lateral flow and also illuminate the wide application the vertical flow assay has been used over the time for e.g. https://doi.org/10.1007/s10544-020-00480-w 

In figure 2, text seem to be falling out, kindly correct it.

Mention the significance of error bar in fig 3 

In figure 6 A, what does the last set of NCM represent, the legend information is missing.  

Overall figure resolution needs to be updated throughout the manuscript

Author Response

(1) A complete spell check needs to be done

Authors: We have run a spell check on the document. Some words, such as Cytiva, appear misspelled, but they are company names.

(2) Also, there is inconsistency in figure number, kindly update it.

Authors: Thank you for pointing this out. We have corrected the figure numbering.

(3) Manuscript needs to be checked for grammatical errors.

Authors: We have checked the manuscript and fixed the errors.

(4) In introduction section author could cite addition reference showing advantage of vertical flow assay over lateral flow and also illuminate the wide application the vertical flow assay has been used over the time for e.g. https://doi.org/10.1007/s10544-020-00480-w

Authors: Thank you for providing the additional reference, we have added it into the introduction.

(5) In figure 2, text seem to be falling out, kindly correct it.

Authors: Thank you, we have fixed the text.

(6) Mention the significance of error bar in fig 3

Authors: We have mentioned the significance of the error bars in the legend of figure 3.

(7) In figure 6 A, what does the last set of NCM represent, the legend information is missing. 

Authors: Our apologies, the label spacing was incorrect so the concentrations did not align with the NCM. We have corrected this.

(8) Overall figure resolution needs to be updated throughout the manuscript

Authors: We thank the reviewer for the suggestion. All figures have been converted to 600 dpi.

Reviewer 3 Report

The manuscript has been improved after revision. However,  redundancy in writing needs to be checked and improved throughout the manuscript (Eg: line no-74-81).

Author Response

The manuscript has been improved after revision. However,  redundancy in writing needs to be checked and improved throughout the manuscript (Eg: line no-74-81).

Author: We thank the reviewer for the suggestion. We have reduced the redundancy within the text.